# Mind the Gap: Animal Protection Law and Opinion of Sheep Farmers and Lay Citizens Regarding Animal Maltreatment in Sheep Farming in Southern Brazil

**DOI:** 10.3390/ani11071903

**Published:** 2021-06-26

**Authors:** Vanessa Souza Soriano, Clive Julian Christie Phillips, Cesar Augusto Taconeli, Alessandra Akemi Hashimoto Fragoso, Carla Forte Maiolino Molento

**Affiliations:** 1Animal Welfare Laboratory, Federal University of Paraná, Rua dos Funcionários, 1540, Curitiba 80035-050, Paraná, Brazil; vanessasszoot@gmail.com (V.S.S.); clive.phillips58@outlook.com (C.J.C.P.); hashimoto@ufpr.br (A.A.H.F.); 2Sustainable Policy Institute, Curtin University, Kent St., Bentley, WA 6102, Australia; 3Department of Statistics, Federal University of Paraná, Rua Cel. Francisco Heráclito dos Santos, 100, Curitiba 81531-980, Paraná, Brazil; cetaconeli@gmail.com

**Keywords:** animal abuse, animal protection law, animal welfare, farmers’ opinion, sheep welfare

## Abstract

**Simple Summary:**

Animal protection laws are written and enforced differently depending on the category of animals they are assigned to. This generates inconsistencies in the recognition of animal maltreatment. We studied sheep farmers’ and other citizens’ opinions regarding animal maltreatment by discussing the risk of sheep maltreatment in regular farming practices in Southern Brazil. We surveyed the perception of 56 farmers and 209 citizens regarding general animal and specific on-farm sheep maltreatment issues. The understanding of some key components of animal maltreatment was similar for both respondent groups: failing to provide for the basic animal needs and aggression or physical abuse towards animals. However, citizens were more sensitive than farmers to animal stress, suffering, fear and pain. More citizens than sheep farmers believed that animal maltreatment occurs in sheep farming; nevertheless, nearly half of the farmers recognized sheep maltreatment within normal production practices. Most citizens and all of the farmers were unaware of the Brazilian animal protection laws. Most citizens stated that they would not purchase products from animals exposed to maltreatment. We suggest painful procedures as the main risk of animal maltreatment in sheep farming and a priority issue. The level of cognitive dissonance in sheep farmers and contradictions between farmers and other citizens observed in our results indicate that mitigation policies are urgently needed.

**Abstract:**

We aimed to study the gaps between the law and sheep farmer and citizen opinions regarding animal maltreatment by discussing the risk of sheep maltreatment in regular farming practices in Southern Brazil. We surveyed the perception of 56 farmers and 209 citizens regarding general animal and specific on-farm sheep maltreatment issues. The main themes from these two groups about the key components of animal maltreatment were similar: failing to provide for the basic animal needs (27.0%; 96 of 355 total quotes) and aggression or physical abuse (23.9%; 85/355). However, citizens (19.8%; 60/303) were more sensitive than farmers (9.6%; 5/52) to animal stress, suffering, fear, pain or painful procedures (*p* < 0.05). The perspective of citizens was closer than that of farmers to expert definitions for three situations: emaciation, movement restriction and tail docking without anesthetic use (*p* < 0.05). More citizens (71.6%; 116/162) than sheep farmers (49.0%; 24/49) believed that animal maltreatment occurs in sheep farming (*p* < 0.05), but nearly half of the farmers recognized sheep maltreatment within regular production practices. Most citizens (86.4%; 140/162) and all farmers (100.0%; 0/51) were unaware of any Brazilian animal protection law. Most citizens (79%; 131/167) stated that they would not purchase products from animals exposed to maltreatment. We suggest painful procedures as a major risk of animal maltreatment in sheep farming and a priority issue. With the many decades of animal protection laws and scientific recognition of animal sentience and welfare requirements, the level of cognitive dissonance and practical contradictions observed in our results indicate that mitigation policies are urgently needed.

## 1. Introduction

Questions regarding the ethics of farm animal production are being raised in parallel with the pressure for increasing total food production in the world. In 2016, for example, according to the Eurobarometer [1], 94% of European citizens believed that it is important to protect the welfare of farmed animals. For this paper, we refer to animal welfare as the physical and mental state of an animal in relation to the conditions in which it lives and dies, considering that an animal experiences good welfare if the animal is healthy, comfortable, well nourished, safe, is not suffering from unpleasant states such as pain, fear and distress and is able to express behaviors that are important for its physical and mental state [2]. In addition, it seems that the traditional cost–benefit analysis needs to be extended beyond financial implications to include an understanding of the degree of acceptable animal suffering [3]. Even though the speed with which ethical concerns related to animals is growing in each country is variable, society’s demand for better animal welfare in modern animal production seems clear. Due to this increasingly strong public concern related to farm animals, welfare policies have been gradually developed and implemented worldwide, which is an ongoing process. For example, Broom [4] published a list of European Union (EU) animal welfare policy and legislation, registering a period of these developments.

Many European countries have opted to regulate production methods through legislation [5]. In the early to mid-twentieth century, a number of countries developed legislation about the prevention of cruelty to animals and unnecessary animal suffering [6], including farm animals. In Brazil, the main laws protecting animals—Federal Constitution, Article 225 [7] and the Federal Environmental Law 9.605/1998 [8]—prohibit cruelty and acts of abuse and maltreatment against animals with no distinction in terms of animal species. There is only one exception, which is for the scientific use of animals when no alternative is available. Despite the broad scope of this law, the general terms adopted in Brazilian legislation, such as cruelty, abuse and maltreatment, may lead to differences in their interpretation by authorities. Differences in interpretation are evident from current practices, as, for example, guidelines for agricultural activities frequently condone practices that would in other spheres be considered animal maltreatment. These include castrating or tail docking without anesthesia in sheep and pigs, macerating live chicks or plucking an animal’s eye out for female shrimp reproductive management.

Maltreatment can be defined as actions or omissions that are neglectful, abusive or a threat to the welfare of an individual [9]. Neglect is an act of omission involving failure to provide for the basic physical and emotional needs of a dependent being [9,10], and it is the most common form of animal maltreatment [11]. Recognition of negligence as a form of maltreatment requires special attention because it is less obvious than physical aggression [10]. In contrast to the passive nature of neglect, abuse is an active process consisting of acts of aggression with intent to physically or emotionally harm the victim [10], and it may include acts of commission or omission while conscious of the fact that the result will inflict harm [9]. The Brazilian Protocol for Expert Report on Animal Welfare (PERAW) was designed to identify animal maltreatment situations and to contribute to court decisions regarding crimes against animals [12]. PERAW was devised to deal with suspicion of cruelty towards dogs, cats and horses [13]; however, its structure allows for adaptation to different animal species and scenarios [12], and recently, it was adapted for extensively farmed sheep.

When considering contemporary farming systems, animal protection legislation suggests a vulnerability for animals and producers, as it may conflict with routine procedures in animal production systems, which, in turn, suggests both the vulnerability of animals suffering from these procedures and of farmers who become liable to prosecution. The context also involves market vulnerability since consumer attitudes toward a company may be associated with its social license to operate. Consumer perception that a company is socially oriented is associated with a higher level of trust in the company and its products, influencing consumer actions [14]. In the case of farming systems, there are numerous videos showing abusive handling, and consumers often become more concerned about animal welfare after a shocking undercover video is released [15,16]. As an example, in January 2008, an undercover investigation captured video footage of workers in a United States (US) company dragging, kicking and administering electrical shocks to dairy cows that were unable to walk, leading to the largest meat recall in history with more than 64 million kg of beef, a multi-million-dollar civil lawsuit and multiple felony animal cruelty convictions [17]. Thus, online media’s power to convey information in general and, more specifically, on animal maltreatment is extremely high, with increasing chances for transparency and exposure of on-field situations that may impact consumer choices. This may be associated with the fact that society has been changing the limits of acceptability in terms of animal protection, which according to Thompson et al. [18], constitutes an increasingly rapid cultural change.

Farmers must frequently make a trade-off between economic interests and animal welfare-related concerns [19]. However, farmers and non-farming citizens (hereafter citizens) might have different opinions as to what they consider to be animal maltreatment in farming systems. Animal welfare is considered more important by the general public than farmers, scientists and service providers because of the vested interests of these stakeholders in the industry [20]. Thus, animal maltreatment in farming is a sensitive issue; however, it is essential to understand the contextual basis for opinions if improvements for citizens, farmers and animals are to be achieved. We aimed to study the gaps between the law and sheep farmer and citizen opinions regarding sheep maltreatment in Southern Brazil.

## 2. Materials and Methods

This study was designed in two stages. First, we studied the opinions of farmers and citizens regarding animal maltreatment generally and sheep maltreatment on-farm specifically. Then we used the PERAW, adapted for sheep, as the technical description of the situations regarding animal maltreatment in relation to Brazilian animal protection laws. This research was approved by the Ethics Committee of Research with Human Beings of the Federal University of Paraná, Brazil (registration number 2.248.306/2017).

### Animal Maltreatment: Opinion of Sheep Farmers and Lay Citizens

The survey to elicit opinions related to sheep farming and the opinion of farmers and citizens regarding animal maltreatment was conducted with respondents from the State of Rio Grande do Sul, Brazil, from September to December 2017. Sheep farmers were from a southwest town of the State and citizens from the metropolitan region of the State capital, Porto Alegre. Questionnaires were semi-structured with open-ended and multiple-choice questions, with 20 questions for farmers and 17 questions for citizens. Questions were of three types [21]: (1) respondents’ characteristics (demographic), (2) farm characteristics, events and management of sheep farming and (3) perception or attitudes toward animal maltreatment. Each question addressed one or more of these topics (Table 1).

To compare the opinion of farmers and citizens about animal maltreatment, we asked one open-ended and seven multiple-choice questions (Table 1). Considering the multiple-choice questions, four situations were representative of each group of indicators reported by Molento and Hammerschmidt [12] as animal maltreatment: an emaciated animal as an example of nutritional maltreatment, movement restriction as comfort maltreatment, a diseased and untreated animal as health maltreatment, and an isolated animal as behavioral maltreatment; the situation “tail docking without anesthetic use” was also presented as an example of a routine invasive procedure in sheep production that is classified as animal maltreatment according to Brazilian legislation [8,22].

Because not all farmers had access to the internet, data collection included direct interviews and online questionnaires; for citizens, all participants responded to an online questionnaire. Online questionnaires were created and hosted in an online platform (https://www.onlinepesquisa.com accessed on 26 June 2021). The questionnaire to sheep farmers was pilot-tested with three veterinarians and four sheep farmers to ensure that questions allowed for correct interpretation by the participants and thus yielding relevant responses. The same pilot test was performed for the citizens’ questionnaire, administered to two citizens and two veterinarians. No changes in the questionnaires were made.

To distribute the online version of the questionnaire to sheep farmers, we created a standard text including a link to the questionnaire, inviting sheep farmers to both participate themselves and to encourage other sheep farmers to complete the questionnaire, employing a purposive sampling technique [23]. The standard text was sent by email to a list of 20 farmers in the Sindicato Rural de Quaraí (Rural Syndicate of Quaraí). Additionally, we sent the standard text by the applicative ‘WhatsApp’ (WhatsApp LLC, Santa Clara, CA, USA) on cell phones to two Whatsapp sheep farmer groups, including 51 and 35 participants. In addition, we contacted sheep farmers by telephone and asked if they were willing to participate by providing their emails to receive the questionnaire link. Copies of the questionnaire on paper were also provided to the Rural Syndicate, Wool Cooperative, Municipal Agriculture Secretariat, agricultural shops and community members.

To invite citizens, we created a social media page on Facebook, presenting the questionnaire link. In addition, we created an email address for this study and created a standard email including the link to the questionnaire, inviting people to participate and to spread it to others. We sent the standard email text to graduation program contacts in universities, involving a variety of programs: pharmacology, physics, nutrition, engineering, veterinary medicine, biology, nursing, statistics and sociology. Additionally, we searched for public citizen email addresses on Google, with keywords regarding employments such as doctor, therapist, math, environmental, cultural, restaurant, floriculture, barbecue, philosopher, taxi, art, religion, massage therapist, manicure, home services, office boys, yoga, meat, nurse, lawyers, student, language professor, personal trainer, coffee store, landlord, army, chefs, retired, decorator, engineer, human resources and photographer. The same standard email was sent to all addresses obtained. In these cases, we considered from six to ten pages of results from Google for each key word, resulting in an overall number of 2560 messages sent.

From 280 sheep farmers registered by the Municipal Secretary of Agriculture (MSU) [24], 56 participated: 18 by online link and 38 by paper questionnaire. According to Brazilian Institute of Geography and Statistics (IBGE) [25], most sheep farmers in this region have a low level of education, and many have no access to the internet or telephone in rural areas. The overall level of education in the town was 54.8% of people aged 10 or more years had no instruction or incomplete elementary school [25]. Of the 56 farmers in this study, 73% (41) were men. Sheep farming was not the main activity on 79% (44/56) of the farms, and the mean flock size was 477 (25 min and 2600 max) sheep. In terms of sheep farming systems, 80% (45/56) of farmers raised sheep on native and cultivated pastures; 18% (10/56) raised sheep on native pastures alone; 2% (1/56) raised sheep on native pasture during daytime and housed them at night.

Farm management characteristics are presented in Table 2. During the descriptive section, one farmer left the questionnaire.

A total of 220 citizens answered the survey, of which 209 fulfilled the criterion of living in the metropolitan region of Porto Alegre and age > 18 years. Of all respondents, 70% (147/209) were female, 83% (173/209) were graduates or post-graduates, and 93% (194/209) ate meat frequently or occasionally on the weekends (Table 3). According to the last Brazilian demographic census [24], 51.3% of the state population are female and only 11.3% of people aged 25 years or more are graduates or post-graduates. In terms of employment, we classified respondents according to their contact with animals at work: 11% of citizens had potential contact with animals, including veterinarians, veterinarian assistants, biologists, animal scientists and environmental engineers; 35% (71/205) were in activities with no contact with animals such as a doctor, nurse, federal judge, lawyer, bank clerk, office assistant, massage therapist, pilot, tour guide, physicist, nutritionist, civil engineer, computer system analyst, housewife, manicure, etc.; 54% (111/205) did not detail their activities regarding experience or contact with animals (Table 3). In these cases, answers given were professor, researcher, student, retired, consulter and public servant.

Open-ended questions were analyzed by the Discourse of the Collective Subject method [26], which determines a collective opinion by extracting central ideas from individual respondent statements. This methodology facilitates the evaluation of open-ended questions by a qualitative and quantitative representation of a group of individuals [27]. For each open-ended question, we identified central ideas and classified all arguments or quotes that matched the same central idea; each response may represent more than one central idea. As an example, for the question about animal maltreatment definition, we classified answers according to the following central ideas: welfare-significant internal states as “nutrition, environment and health,” welfare-significant external circumstances relating to behavior as “space restriction or animal containment,” “lack of naturalness or isolation,” or “physical aggression/ abuse,” and mental state as “stress/suffering/fear/pain/painful procedures,” or “emotional neglect,” or “abandonment” [28]; answers that did not fit into the previous central ideas were considered as “non-classifiable.”

The prevalence of central ideas given in response to the question regarding the definition of animal maltreatment was compared by the chi-square test. The associations between the type of respondent and the five-point Likert scale opinions regarding situations understood to include animal maltreatment, the occurrence of animal maltreatment on sheep farming and the knowledge of laws regarding animal maltreatment were evaluated by the Cochran–Armitage trend test. When the expected frequencies under the null hypothesis of no association were less than five, simulated *p*-values were obtained based on 5000 simulations.

Latent Class Analysis (LCA) [29] was used for answers to questions asked of both farmers and citizens. The main goal of using LCA in this study was to identify different patterns of respondents based on their corresponding answers. We considered two groups of data: (A) central ideas answered to the open question regarding the different opinions about the definition of animal maltreatment definition; (B) the seven questions answered using a five-point Likert scale: the opinion regarding situations including (1) an emaciated animal, (2) movement restriction, (3) a diseased and untreated animal, (4) an isolated animal, (5) an animal whose tail was docked without anesthetic use; (6) opinions about animal maltreatment occurrence on sheep farms and (7) the knowledge of laws regarding animal maltreatment. The first step of the LCA was to select the number of latent classes present in the sample. We investigated solutions varying from one to four latent classes and selected the number of latent classes as one that minimized the Bayesian Information Criterion (BIC), which configures a measure that weights the model goodness of fit and its complexity. Analyses were performed using R Statistical Computing Environment version 3.6.0 (R Foundation for Statistical Computing, Vienna, Austria).

To identify the risk of animal maltreatment in sheep, we used the indicators presented in the Animal Welfare Indicators (AWIN) protocol for sheep [30] and the structure of the PERAW [12] as a guide for the evaluation of four indicator groups: (1) nutritional indicators, (2) comfort indicators, (3) health indicators and (4) behavioral indicators.

## 3. Results

### 3.1. Opinion of Sheep Farmers and Lay Citizens Regarding Animal Maltreatment in General

Answers to the question on what constitutes animal maltreatment (Table 4) by 65.4% (36/55) of farmers and 66.5% (139/209) of citizens cited the same two main ideas: failure to provide for the basic animal needs (27.0%; 96 of 355 total quotes) and aggression or physical abuse (23.9%; 85/355). The third most cited idea by 19.8% (60/303) citizens, and it was differently of 9.6% (5/52) farmers (*p* < 0.05), referred to animal stress or suffering or fear or pain or painful procedures. Deviation from naturalness (9.6%; 5/52) and non-classifiable ideas (9.6%; 5/52) shared third place in the frequency of farmer citations and did not differ in their citation frequency between citizens and farmers (*p* > 0.05). Perceptions about animal maltreatment in terms of space restriction or animal confinement and abandonment were most cited by citizens (*p* < 0.05).

Two latent classes were detected for the eight main ideas cited by respondents as animal maltreatment definition, as the optimal number for both groups of data, providing the lowest BIC in both cases. The sample size used to analyze the variables in group B was smaller than A because respondents failed to answer some of the eight questions. These latent classes were compared through the conditional probabilities of expressing each central idea, as can be seen in Figure 1. Based on these conditional probabilities, the latent classes produced are: LC1—associated with the ideas of failing to provide for basic animal needs, aggression or physical abuse and movement restriction; LC2—associated with the ideas of animal stress, or suffering, or fear, or pain, or painful procedures and aggression or physical abuse; χ2 = 4,14 and *p* = 0.04 (Figure 1). Following this, each farmer was classified into one of the LC by allocating them to the LC that presents the highest conditional probability. The LC1 was composed of responses of 40 farmers (72.7%) and 118 citizens (56.5%) and the latent class 2 by responses of 15 farmers (27.3%) and 91 citizens (43.5%). Additionally, the probability of perceiving the ideas in LC1 as animal maltreatment was higher than regarding the ideas in LC2.

### 3.2. Opinion of Sheep Farmers and Citizens about Animal Maltreatment in Specific Situations, and Their Knowledge of Brazilian Laws Regarding Animal Protection

Most farmers and citizens considered the five situations presented to them as animal maltreatment (Figure 2). Citizens were more sensitive than farmers for three situations: “an emaciated animal,” “tail docking without anesthetic use” and “an animal in a local with movement restriction” (Figure 2; *p* < 0.05). The order of situations perceived as maltreatment by farmers was “a diseased and untreated animal” > “an animal social isolated and with no contact to others animals” > “an animal in a local with movement restriction” > “an emaciated animal” > “tail docking without anesthetic uses”; the order perceived by citizens was “an animal social isolated and with no contact to others animals” > “a diseased and untreated animal” > “tail docking without anesthetic uses” > “an animal in a local with movement restriction” > “an emaciated animal” (Figure 2). Animal maltreatment occurrence in sheep farming was perceived differently by farmers and citizens (Figure 2; *p* < 0.05); citizens were more sensitive than farmers: most citizens (71.6%; 116/162) believed that animal maltreatment occurs in sheep farming, and 50.6% (82/162) of citizens and 32.7% (16/49) of farmers answered “definitely yes” to animal maltreatment occurrence in sheep farming (Figure 2).

Most respondents, and especially the farmers, did not have knowledge of Brazilian laws regarding animal protection, with 88.2% (45/51) of farmers and 63.0% (102/162) of citizens answering “no” or “definitely no,” and 5.9% (3/51) of farmers and 17.9% (29/162) of citizens answering “yes” or “definitely yes.” Thus, citizens reported more knowledge of Brazilian laws regarding animal protection than farmers (*p* = 0.0017). Of all citizens, 13.6% (22/162) cited at least one law (Figure 3), with two citizens reporting two laws. The most cited (7.4%, 12/162) was the Environmental Law 9.605/1998 [8] (Figure 3). Although 5.9% (3/51) of farmers reported knowing Brazilian laws regarding animal maltreatment, no law was cited by them (Figure 3).

We found two latent classes for the six questions in Figure 2 and the question on law knowledge as the optimal number for both groups of data, providing the lowest BIC in both cases. These can be characterized as: LC1—associated with responses of “no” or “definitely no” to the perception of different situations as animal maltreatment and law knowledge; LC2—associated with responses of “yes” or “definitely yes” to the perception of different situations as animal maltreatment and law knowledge. Based on the individual conditional probabilities, LC1 comprised the responses of 42 farmers (97.7%; 42/43) and eight citizens (5.5%; 8/145), and LC2 by the responses of one farmer (2.3%; 1/43) and 137 citizens (94.5%; 137/145); with χ2 = 144.28 and *p* = 0.0002. The LC2 exhibited a more sensitive perception in terms of animal maltreatment, and it was composed basically by citizens. The Latent Class Analysis showed a clear and different pattern of perspective between farmers and citizens for the multiple-choice questions.

### 3.3. Citizen Attitudes toward Animal Maltreatment

Citizen answers to the questions “Considering a scale from 1 (not relevant) to 5 (very relevant), what is your opinion about the relevance of animal maltreatment debates?” and “Why?” are presented in Table 5. Of 168 respondents, 90% believed that animal maltreatment debates are relevant or very relevant, mainly due to animal ethical issues (50%; 38/76). Other ideas reported for justifying the relevance of animal maltreatment debates were related to the fact that animals are living and sentient beings (34.2%; 26/76) and to understand the context and propose a solution (34.2%; 26/76).

Citizen answers to the question “Considering a scale from 1 (definitely yes) to 5 (definitely no), would your purchase products from animals if you knew they were maltreated while on the farm?” and “Why?” are presented in Table 6. Most respondents (79%; 131/167) stated that they would not purchase products from animals who were maltreated on the farm, and the most frequent reasons given were concerns about animals (45.3%; 44/97) and not to contribute to the patronage of companies using bad practices to animals, as well as boycotting companies using bad practices (24.7%; 24/97). Of 167 respondents, 11% were neutral, mostly due to the lack of options in the market (50%; 4/8) and for cultural reasons (37.5%; 3/8).

### 3.4. Protocol for Expert Report on Animal Welfare for Sheep: Expert Perspective 

There are 23 indicators in AWIN for sheep [30]: (1) body condition, (2) lamb mortality and (3) water availability in the nutrition group of indicators; (1) panting, (2) access to shade or shelter, (3) fleece cleanliness, (4) stocking density and (5) hoof overgrowth in the comfort group of indicators; (1) body and head lesions, (2) leg injuries, (3) mucosa color, (4) ocular discharge, (5) respiratory quality, (6) lameness, (7) fleece quality, (8) fecal soiling, (9) mastitis and udder lesion and (10) tail length in the health group of indicators; and (1) social withdrawal, (2) excessive itching, (3) familiar human approach and (4) qualitative behavior assessment in the behavior group of indicators. From these, we considered: body condition to relate to the opinion of respondents regarding “an emaciated animal,” stocking density to relate to the opinion of respondents regarding “an animal with movement restriction,” tail length to relate to the opinion of respondents regarding “tail docking without anesthetic use” and social withdrawal to relate to the opinion of respondents regarding “an isolated animal.” Additionally, to discuss the opinion of respondents to “a diseased and untreated animal,” as well as all other potential maltreatment situations, we considered the “inadequate” classification of welfare indicator groups according to PERAW [12]. The expert perspective to discuss the answers of farmers and citizens related to animal maltreatment in sheep farming was based on the overall conclusion in the PERAW.

## 4. Discussion

In general, citizens showed a more sensitive perception of animal maltreatment than farmers, and our results support previous evidence that animal maltreatment is a serious societal concern.

### 4.1. Perception of Sheep Farmers and Citizens Regarding Animal Welfare

The two main ideas cited by both groups of respondents were similar: aggression or physical abuse and failure to provide for basic animal needs. These results were expected since maltreatment in animals has been traditionally focused on physical harm due to the fact that the outcome of physical trauma is easily visible and can cause death, whereas emotional abuse is not expected to have such obvious and extreme outcomes [9]. In the same way, failure to provide for animal needs, with evidence such as very thin animals, for example, may be visible [31] and considered by respondents as a threat to the animals’ lives. This is classified as neglect [9], which is the most prevalent cause of animal maltreatment [10]. According to Mellor [28], hunger impels animals to engage in behaviors that help to secure their survival. Thus, it seems that respondents linked animal maltreatment mainly to survival and physical threat aspects when spontaneously answering the open-ended questions.

The third idea most cited by citizens as animal maltreatment was animal stress, suffering, fear, pain or painful procedures, and it was different from the farmers. This idea is associated with negative mental states in response to physical harm or threat. Instillation of fear, anxiety, anguish and social isolation was considered to be mental abuse by Patronek [31]. Additionally, in the case of pain, for example, physical abuse can cause emotional maltreatment if it causes emotional distress as a response to the pain [10]. In terms of passive acts, failure to provide relief when pain is detected is also animal neglect [32]. Farmers mentioned the idea of stress, suffering, fear, pain or painful procedures less frequently, probably because they consider fear as a normal sheep reaction to many on-farm situations, such as gathering, shearing and health treatments, as well as pain during diseases and routine procedures. Similarly, Doughty et al. [20] observed that the general public in Australia attributed greater importance than farmers to pain and fear. Painful procedures as castration and tail docking were also mentioned by 98% (54/55) of farmers as usual practice on sheep farming, carried out without anesthetic use (96%; 52/54). Hence, it was expected that most of them did not think of painful procedures as animal maltreatment, considering it normal. In a study with pig farmers, Albernaz-Gonçalvez et al. [33] found that although they recognized pigs as sentient beings, they maintained negative attitudes towards practices that could improve pig welfare or at least minimize pain. Many management and animal-based indicators of poor welfare were observed, such as the use of painful and stressful management practices and use of environments that limit the expression of natural pig behaviors; however, most farmers were satisfied with animal welfare standards on their farms, and not all farmers recognized the pain caused by their routine practices [33]. The comparison of the perceptions of suffering expressed by sheep (our results) and pig [33] farmers suggests that the banalization of animal suffering may be more present for those involved in the intensified systems, a hypothesis that warrants further studies.

Other ideas cited more often by citizens than farmers as animal maltreatment were movement restriction and abandonment. Lack of sufficient living space, termed movement restriction in this study, was considered to be emotional neglect, and abandonment is described as emotional abuse, as the categories of animal maltreatment described by Merck [10]. Animals who cannot move around freely, those kept in very small cages that prevent them from performing most of their behaviors and an unnatural life are the things that most disturb people [34]. Farmers may get used to the idea of animal movement restriction because they are frequently immersed in production systems where animals are raised in small spaces. Additionally, abandonment was not cited by farmers; this suggests that farmers and citizens may have had different animal species and contexts in mind while responding to this first part of the questionnaire. Because dog abandonment is a common cause of urban animal maltreatment allegations in Brazil [11], it is likely that more citizens than farmers thought of pet animals when answering the survey.

Two latent classes of respondents regarding their probabilities to describe the main specific ideas as animal maltreatment were observed. One pattern was formed by 59.8% (158/264, 73% of farmers) of respondents who associated animal maltreatment mostly with negative physical conditions, such as failure to provide for basic animal needs, aggression or physical abuse, and movement restriction, which may be correlated to Merck’s [10] categories of physical neglect, physical abuse and emotional neglect, respectively. The other pattern was composed of 40.2% (106/264, 27% of farmers) of respondents, who linked animal maltreatment mostly to negative mental states such as animal stress, suffering, fear, pain or painful procedures, and with physical harm such as aggression or physical abuse, which, in turn, are associated to Merck’s [10] categories of emotional neglect and abuse. Thus, while the physical nature of animal maltreatment was the most evident aspect reported by both patterns of respondents, the emotional aspect was presented as an important factor as well. Emotional maltreatment refers to the link between emotional states and physical health [10], and our results provided evidence that emotional maltreatment, even when not clearly physically manifested, is relevant to society.

### 4.2. Protocol for Expert Report on Animal Welfare (PERAW) and the Perception of Animal Maltreatment by Sheep Farmers and Lay Citizens Regarding Specific Issues

In the latent class analysis, citizens presented more sensitive responses than farmers regarding the perception of animal maltreatment in specific issues, which was probably related to the higher importance to animal welfare given by citizens than farmers [20]. The perspective of citizens was closer than that of farmers to the technical results given by PERAW for three of the five situations studied: “an emaciated animal,” “an animal in a local with movement restriction” and “tail docking without anesthetic use.”

#### 4.2.1. Nutrition Group of Indicators

Most respondents considered emaciated animals as an animal maltreatment situation, in accordance with PERAW. It was expected that an emaciated animal would be considered animal maltreatment since hunger is a basic need and, according to Mellor [28], a survival-related indicator. Thus, the undoubted negativity of hunger creates a sense of urgency to respond. The reduced perception of farmers in relation to that of citizens of an emaciated animal as a maltreatment situation may be related to the fact that farmers considered body condition variation in sheep farming, including emaciation, as normal. For instance, in one study of 442 ewes, 62% had a low body condition score of between 1 and 2.5 [35]. Additionally, the fact that sheep farming was not the main activity of the farms probably means that sheep were not raised on the best fields because they are less profitable than cattle. In a similar study to ours, a higher mean value was given for perceived importance of feed availability to farm animals by citizens, scoring 4.2 on a five-point scale, than by farmers, scoring 4.0 [36].

#### 4.2.2. Comfort Group of Indicators

The comfort group was mainly to do with movement restrictions, and citizens were more sensitive than farmers to this condition. Similarly, Vanhonacker et al. [19] observed that the Flemish societal concern for both stocking density and pen size were most important within the full picture of farm animal welfare. In another study, the importance of available space was scored higher (4.2 on a five-point scale) by citizens than farmers (3.7) [36]. Different views may again be related to a banalizing effect of constant exposure of farmers to routine farm animal practices. Most citizens, however, are not desensitized. For example, Franco et al. [37] observed that 89.9% of Brazilian citizens reported knowing little or nothing about how farm animals are raised. Additionally, Hötzel et al. [38] found that 86.1% of Brazilian citizens rejected zero-grazing for dairy cows after receiving a short statement about the practice. Additionally, although the survey presented questions on farm animals before the animal maltreatment specific conditions, it is possible that some citizens answered this question thinking about pets confined in small places. Some farmers may have considered movement restriction acceptable because more extreme restriction occurs in other livestock production systems, such as cages for hens and individual stalls for sows. Extensively managed sheep are housed in specific situations such as lambing or during disease recovery only for a short period, including when used for routine management, such as vaccination. Thus, it is possible that farmers were less sensitive than citizens to movement restriction as they may have linked it with beneficial situations for the animals within the specific production context that sheep are inserted.

#### 4.2.3. Health Group of Indicators

The health group of the indicators, presented as “a diseased and untreated animal,” was the situation most cited as animal maltreatment by the farmers and the second by citizens. This was expected since a disease is an evident threat to the animals’ life and performance. Additionally, it is in the farmers’ interests to preserve elements of farm animal welfare related to good animal health [39]. Animals experiencing disease but not offered veterinary assistance is an inadequate situation [12], and thus respondents’ perception was in accordance with expert opinion. Animal health also received high importance scores in Vanhonacker et al.’s [36] research: from 72 aspects, citizens reported “disease” as the third most important factor (4.37, using a 1–5 scale) and farmers the fifth (4.13). However, we did not observe a significant difference between the opinion of citizens and farmers for importance attributed to diseases.

Tail docking without anesthetic use was perceived as an animal maltreatment situation by almost half of the farmers and by most citizens, whose views were statistically different. The perception of citizens was closer to expert reports than the perception of farmers. Similarly, using a scale from 1 = no compromise to welfare to 5 = significantly compromises welfare, [20] found that the public in Australia scored husbandry practices (3.83) as more important than did sheep farmers (2.73). The higher perception of animal maltreatment by citizens may be related to the fact that painful procedures performed on animals are among the most emotive of public concerns about animal welfare [40]. The lower perception of farmers was not a surprise since 96% (52/54) of them reported not using anesthetic during castration and tail docking procedures. Importantly, our results (that most farmers performed tail docking without anesthesia and that almost half of them consider it to be animal maltreatment) suggest that almost half of the farmers were in cognitive dissonance, demonstrating conflicts between beliefs and behaviors as described by the theory of Festinger [41].

James and Hendrickson [42] found that farmers reported higher levels of concern when ethical issues involved harm to the animals. Because most farmers docked sheep’s tails in sheep farming (98%; 54/55), it is possible that some of them do not perceive this procedure as harmful to the animal. In addition, for farmers, animal suffering is part of daily life and of producing food to feed society [43]. Another explanation may be related to the belief amongst sheep farmers that tail docking improves mating and farm sanitary conditions [44], beliefs not supported by scientific evidence. In addition, due to the tradition of docking sheep tails, farmers usually transfer this practice across generations. According to Curnow et al. [45], sheep farmers need to understand, test and trust the value of innovation before it can be successfully applied on farms.

Docking sheep tails without anesthesia is illegal according to Brazilian law 9.605/98 [8] and Resolution 877/08 of the Brazilian Federal Veterinary Council [22]. When anesthetics are used, and thus, pain is controlled, short-docked tails are still an inadequate procedure due to the greater incidences of rectal prolapse [46], predisposition to prolapse of the vagina and cervix in pregnant ewes [47] and the absence of clear scientific evidence of any benefit resulting from sheep tail docking [48].

#### 4.2.4. Behavior Group of Indicators

Most sheep farmers and citizens considered an isolated animal as an animal maltreatment situation. It was perceived as the worst situation when considering the answers classified as “definitely animal maltreatment”. Our results thus demonstrated the high importance for social behavior given by respondents and agreement between respondents’ view and expert perspective. Social isolation is considered as mental abuse by Patronek [31], and the social withdrawal is unnatural for sheep, except when in quarantine or at lambing. In contrast to our results, a higher perceived importance for social behavior in citizens (3.84 in a five-point scale) in comparison to farmers (3.21) was evident in the study of Vanhonacker et al. [36]; however, farmers were from Flanders where sheep farming is not usual.

Social isolation was indicated by Forkman et al. [49] as one of the most stressful components tested for measuring fear and sheep behavioral responses in social isolation, including higher locomotor activity, vocalizations and defecation/urination. Thus, the sensitive perception of farmers may be due to their experiences with sheep showing an evident stressful reaction when isolated. In addition, sheep isolation is normally associated with a purpose such as a health treatment. It is recommended to separate a diseased animal from the flock, especially for infectious diseases such as foot rot [50]. Hence, farmers may associate an isolated animal with an animal that is also sick.

### 4.3. The Risk of Animal Maltreatment in Sheep Farm

Most citizens believed that animal maltreatment occurs in sheep farming. In fact, to identify animal maltreatment in sheep farming using PERAW, at least one group of indicators is needed that presents a situation considered to be inadequate and no human intervention to mitigate the negative consequences for the animal. Most aspects of a sheep’s life are dependent on human decisions, which may have played a role in the perception of citizens. In this way, Tiplady et al. [51] argued that direct human maltreatment of animals received a greater response than other equally harmful events for animals in Australia that were perceived as less directly related to human decisions, such as cattle drowning because of floods or dying during droughts. Thus, the current rapid sharing of footage or videos via social media may intensify exposure of farmers, especially considering the public views on what is unacceptable towards animals. The public in this study perceived that failing to provide for basic animal needs, as in poor nutrition, a bad environment and no health assistance, aggression and physical abuse, and animal stress or suffering or fear or pain or painful procedures occur in sheep farming and are unacceptable. Our results demonstrate that farmers are in a vulnerable position regarding the risks of animal maltreatment charges.

An unexpected result was that almost half of the farmers (49.0%, 24/49) perceived animal maltreatment to occur in sheep farming; 32.7% (16/49) answered as definitely yes, it occurs. This perception raised some issues. First, it reinforces the theory of farmer’s cognitive dissonance since they are responsible for their sheep, and yet, they stated that there is maltreatment against the animals. The cognitive dissonance of dairy farmers was reported by Kristensen and Jakobsen [52] when farmers were confronted with advice suggesting a change of behavior on herd health management. The authors argued that to solve such dissonance, farmers may either comply with the advice or reduce the dissonance by convincing themselves that the suggested change in management is impossible to implement. In these terms, some farmers in this study may think that animal maltreatment situations are inherent in farming, and there are no better options to manage sheep. A second issue is related to the One Welfare concept, which recognizes the interconnections between animal welfare, human wellbeing and the environment, helping to describe the context, to deepen our understanding of the factors involved, and to create a holistic and solution-oriented approach to health and welfare issues [53]. Considering the one welfare concept and farmers’ answers in this study, sheep maltreatment may be associated with farmer personal problems. Studying cases involving pressed charges, Andrade and Anneberg [43] found that Danish farmers who have the highest risk of being convicted of animal neglect are more likely to be troubled by both economic and psychiatric problems, suggesting a link between farm animal neglect and social problems. A third issue is related to economic pressures for improving farm income despite moral costs. James and Hendrickson [42] found evidence that economic pressures result in a greater willingness of farmers to tolerate unethical conduct, particularly in the case of actions that have the potential of causing harm. Overall, the perception farmers expressed in this study suggests that they face an uncomfortable position in relation to some situations in sheep farming.

Our results indicated that most respondents and, particularly, farmers did not know the relevant legislation related to animals, highlighting the vulnerability of farm animals and farmers. If animal protection laws are unknown, it is unlikely that they generate any protection. For farmers, considering that most citizens in this study perceived the occurrence of animal maltreatment in sheep farming and that some of them knew the legislation, farmers may be at risk of formal complaints based on legislation the farmers were not aware of.

According to the Brazilian Law 9.605/1998, Article 32, it is a crime to practice acts of abuse, to maltreat, to hurt or to mutilate wild, domestic or domesticated, either native or exotic animals. Therefore, we believe that the most direct risk of animal maltreatment is related to the health aspect due to tail length inadequacy since tail docking is mutilation and thus explicitly cited as a crime by Brazilian legislation [8]. The other sheep welfare indicators, which are not specifically cited on legislation, depend on an interpretation from experts. Given that there is ample scientific evidence of maltreatment in sheep farming, it is clear that animals and farmers are in a vulnerable situation. Consequently, strategies to improve this situation are urgently needed.

### 4.4. Policy Implications

Legislation is the main policy approach for protecting farm animals, and policy-makers need to be assured that: (i) it will actually improve the welfare of the farm animals concerned; (ii) there is general public support for it; and (iii) it is politically feasible and will not have serious negative economic consequences, for example, on competitiveness and trade [54]. We identified citizen support for preventing on-farm animal maltreatment and an important gap between farmer and citizen opinions, which suggests that policy development is required. A similar disparity between the written law and the realities of the animal law in the Australian context was reported by Morton et al. [55], suggesting that the recognition of a gap in the approaches to animal maltreatment is growing in other countries. A concept that may shed more light in this discussion is agricultural exceptionalism, which has been described as the persistent insulation of agricultural activities from regulations advancing a range of social priorities, not only in the field of animal protection but also in trade, environmental protection and labor and employment law [56].

Changes in farming practices will not necessarily mean negative economic consequences for farmers, as in the case of the unknown tail docking effect on sheep performance; moreover, changes may result in benefits for all stakeholders. However, the traditional practices need questioning and new studies addressing issues that are relevant for farmers and animals. New policies may require field technicians to provide warning to farmers of violation of norms regulating good practices and animal protection, with the goal of increasing advice for farmers to ensure animal protection law compliance. Some pig farmers have indicated that financial incentives for quality of production may be an important motivator or even a requirement for them to make improvements on-farm that benefit animal welfare [33]. Investigation of these motivating factors is warranted for sheep farmers. Our results suggest that information regarding animal protection legislation, especially to farmers, is urgent. Thus, there seems to be room for government action, establishing working groups to propose strategies to reduce the risk of animal maltreatment in farming practices, protecting both farmers and animals. In addition, policies that increase transparency amongst all stakeholders, considering certification schemes to food products and more clarity on labels, for example, will improve consumer understanding of on-farm animal welfare issues so that farmers’ and citizens’ opinions become closer.

### 4.5. Limitations of the Study

There is a degree of bias in the surveyed population, as most citizen respondents were women, a higher percentage compared with the Brazilian demographic census proportion [25], whose greater concern for animals and their welfare as compared to men is well described [20,36]. The fact that more women decided to participate in this study in itself suggests a greater level of interest in the topic, which has also been reported in other research. For instance, Doughty et al. [20] observed that 88% of the general public answering about sheep welfare issues were women, and Ventura et al. [57], when studying contentious practices in dairy farming, observed that 74% of respondents were women. Additionally, women may be at the vanguard of social change on this issue. Farmers in this study possessed some of the characteristics found by Vanhonacker et al. [58] to be associated with very low concern about farm animal welfare: a higher degree of farming experience and a major proportion of rural inhabitants, together with the predominantly male composition of this group of respondents.

Most citizen respondents (82.8%, 173/209) were graduates or postgraduates, a significantly higher proportion compared with the 11.3% of over 24 graduates/postgraduates in the state [25]. Although Doughty et al. [20] did not find significant relationships between education and animal welfare beliefs, Clark et al. [59] reported that those with higher education were likely to be more aware of farm animal welfare issues and tended to be more concerned about modern farming conditions. Therefore, there may be a degree of bias in the citizens surveyed due to their higher educational levels compared to the overall population. Thus, this respondent profile suggests a greater level of interest of citizens about animal maltreatment. This highlights the importance of bringing academic knowledge and research to the citizens by open access to such material and also by publicizing scientific content in less academic formats, such as short videos, podcasts and press releases, amongst others.

In addition to these demographic biases, the ‘snowballing’ technique used to encourage citizens and sheep farmers to participate meant that the responses were not entirely independent, and the influence of some early responders may be considerable.

## 5. Conclusions

This is the first study aiming to understand the differences of opinion of citizens and sheep farmers regarding animal maltreatment. Animal maltreatment was identified as an important issue by citizens, who generally had a negative perception of sheep farming. Although maltreatment is a general term in legislation, experts may detect animal maltreatment in an objective way, and sheep farmers and citizens identified a considerable range of animal maltreatment situations. The most common was physical abuse and neglect. The emotional abuse resultant from pain, stress, fear or painful procedures was also presented as an important factor for citizens and underestimated by farmers. The main risk of animal maltreatment in sheep farming seems to be painful procedures, and thus, they deserve careful consideration to reduce sheep and farmers’ vulnerabilities in the face of citizen expectations and legislation. In addition, minimum planning related to resources on-farm, such as sufficient food availability, is mandatory to avoid inadequate sheep welfare situations related to negligence.

Animal maltreatment in farming systems is a contentious issue; however, all stakeholders play a role in mitigating sheep maltreatment situations. Farmers play the most important role as their decisions directly affect animals; however, it is important for farmers to seek scientific advice, such as recommendations of alternatives to painful procedures on animals. Solutions will reduce farmer and animal vulnerabilities. In addition, more transparency on food products allows consumers to select products with best practices in terms of on-farm animal welfare and may encourage farmers to abolish traditional procedures, such as tail docking. With the many decades of animal protection laws and scientific recognition of animal sentience and welfare requirements, the level of cognitive dissonance and practical contradictions observed in our results indicate that mitigation policies are urgently needed.

## Figures and Tables

**Figure 1 animals-11-01903-f001:**
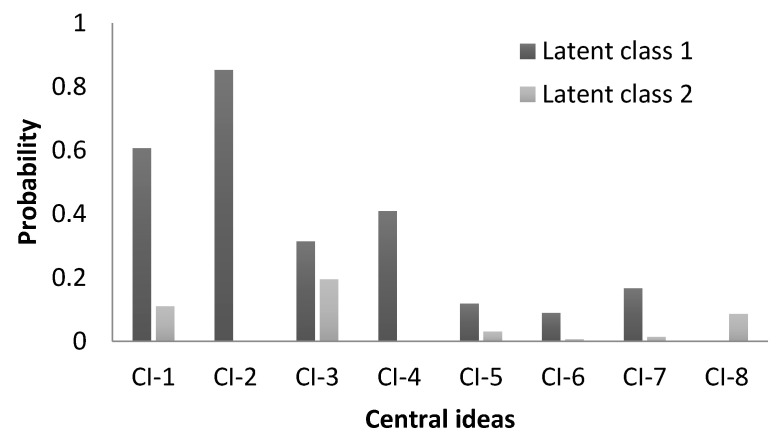
The probability of expressing each central idea (CI) for latent classes 1 and 2, including sheep farmers and citizens in the State of Rio Grande do Sul, Brazil, to consider each central idea as a definition of animal maltreatment: CI-1—Aggression or physical abuse, CI-2—Failure to provide basic needs as good nutrition, good environment or health, CI-3—Stress or suffering or fear or pain or painful procedures, CI-4—Space restriction or animal contained, CI-5—Deviation of naturalness or an isolated animal, CI-6—Emotional neglect, CI-7—Abandonment and CI-8—Non-classifiable.

**Figure 2 animals-11-01903-f002:**
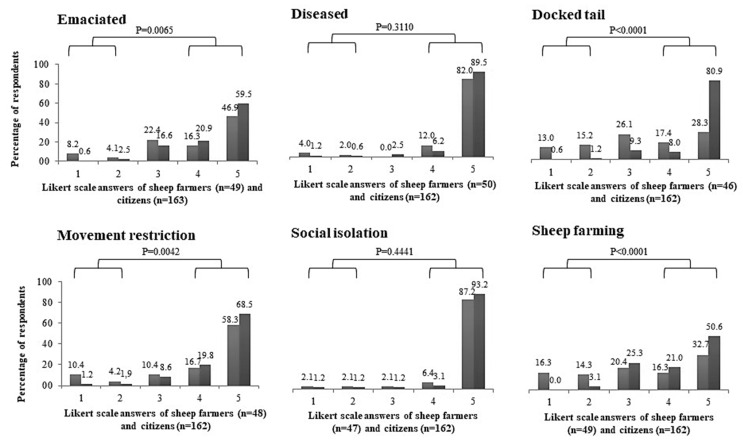
Perception of situations “an emaciated animal,” “a diseased and untreated animal,” “to dock the tail without anesthetic use,” “an animal in a locality with movement restriction,” “an animal socially isolated and with no contact with others animals” as animal maltreatment, and the “perception of animal maltreatment occurrence on sheep farming” on a five-point scale, from 1 (definitely no) to 5 (definitely yes), by sheep farmers (black) in a Southwest town and citizens (gray) in Porto Alegre, Rio Grande do Sul state, Brazil, from September to December 2017; farmer and citizen responses were compared by Cochran–Armitage trend test, in the three categories (Likert 1 + 2, 3, and 4 + 5); *p* < 0.05 indicate a statistical difference.

**Figure 3 animals-11-01903-f003:**
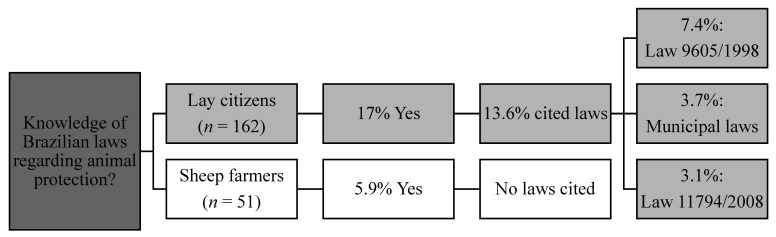
Knowledge of Brazilian laws regarding animal protection of citizens and sheep farmers in the State of Rio Grande do Sul, Brazil, from September to December 2017; Law 9605/1998 lists animal maltreatment as crime and Law 11794/2008 regulates the use of animals in research.

**Table 1 animals-11-01903-t001:** Topics surveyed in a questionnaire for sheep farmers and lay citizens about sheep farming and animal maltreatment in Rio Grande do Sul, Brazil, from September 2017 to December 2017.

Question Type	Group of Respondents	
	Sheep Farmers	Lay Citizens
Demographic	Gender ^1^	Gender ^1^, level of education, employment ^1^, meat consumption ^1,3^
Farm characteristics	Sheep farming as the main activity ^1^, farming system ^1^, flock size ^3^, lamb age at castration ^1,3^, lamb sex and age at tail docking ^1,3^, anesthetic use for castration and tail docking ^1^	-
Perception or attitudes towards animal maltreatment	Perception about animal maltreatment ^3,^*; Opinion regarding situations considered animal maltreatment: emaciated animal, movement restriction, a diseased and untreated animal, isolated animal, tail docking without anesthetic ^2^; Perception of animal maltreatment in sheep farming ^2^; Knowledge of Brazilian laws regarding animal maltreatment ^2,3^	Opinion about the relevance of animal maltreatment debates ^2,3^; Intention to purchase products from animals if they knew animals were in a maltreatment situation on the farm ^2,3^; Perception about animal maltreatment ^3,^*; Opinion regarding situations as animal maltreatment: emaciated animal, movement restriction, a diseased and untreated animal, isolated animal, tail docking without anesthetic ^2^; Perception of animal maltreatment on sheep farming ^2^; Knowledge of Brazilian laws regarding animal maltreatment ^2,3^

^1^ Objective question; ^2^ Five-point scale multiple-choice question; ^3^ Open-ended question (upfront or after “other” option); * Original question was “Currently, it is possible to find news in the media about animal mistreatment, especially against dogs and other pets. What do you understand to be animal maltreatment?”.

**Table 2 animals-11-01903-t002:** The management of sheep castration and tail docking by 56 farmers in a southwest town in the State of Rio Grande do Sul, Brazil, from September 2017 to December 2017.

Variables	Levels	Sheep Farms
Lamb age at castration (*n* = 55)	Does not castrate	1 (2%)
	<one week	2 (4%)
	<two weeks	4 (7%)
	<one month	36 (64%)
	≥one–six months	12 (22%)
Lamb sex and age at tail docking (*n* = 55)	Does not tail-dock	1 (2%)
	Males and females < one month ^1^	26 (36%)
	Males and females > one month ^1^	10 (29%)
	Only females < one month ^1^	11 (20%)
	Only females ≥ one month ^1^	2 (5%)
	Only some females ^2^	1 (2%)
	Males and females	4 (5%)
Anesthetic use in castration and tail docking (*n* = 54)	Yes	2 (4%)
	No	52 (96%)

^1^ Castration and tail docking occur at same time; ^2^ The same farmer who did not castrate lambs.

**Table 3 animals-11-01903-t003:** A description of lay citizen respondents from Porto Alegre, Brazil, in the survey regarding animal welfare and animal maltreatment, from September 2017 to December 2017.

Characteristics	Variables	Respondents	
		Number	%
Gender (*n* = 209)	Male	62	29.7
	Female	147	70.3
Educational level (*n* = 209)	Elementary school	2	1.0
	High school	18	8.6
	Undergraduate degree	16	7.7
	Graduate degree	39	18.7
	Post graduate degree	134	64.1
Employment (*n* = 205)	with animal contact	23	11.2
	with no animal contact	71	34.6
	Information not provided	111	54.1
Meat consumption (*n* = 209)	Yes	188	90.0
	Ocasionally ^1^	6	2.9
	No	15	7.2

^1^ Answers were “rarely,” “on the weekends,” “a little” and “sometimes”.

**Table 4 animals-11-01903-t004:** Central ideas (CI) on what constitutes animal maltreatment by sheep farmers (SF) in a southwestern town in the State of Rio Grande do Sul, and citizens (C, in italic) in Porto Alegre, the State capital, Brazil, from September to December 2017.

Central Ideas for Animal Maltreatment Definition ^1^	Number of Quotes Classified (%)		χ2 Value	*p* Value	Examples of Quotes
	52 SF	*303 C*			
CI 1 Aggression or physical abuse	19 (36.5%)	*66 (21.8%)*	0.18	0.7487	(1)“Violence”; (2) “Slammed the animal” (1) *“Any physical aggression to animal.”* (2) *“… I also consider as maltreatment the use of animals for fun on exhaustive or degrading activities.”*
CI 2 Failure to provide basic needs as nutrition, environment or health	15 (28.8%)	*81 (26.7%)*	2.48	0.1543	(1)“…no health assistance.”; (2) “Hungry or thirsty animals…” (1) *“…not to feed…”* (2) *“...no light, insufficient food, enclosed in spaces with no hygiene.”*
CI 3 Stress or suffering or fear or pain or painful procedures	5 (9.6%)	*60 (19.8%)*	9.03	0.0025	(1) “Practices which impose animal suffering…”; (2) “What makes them feel pain…” (1) *“All that causes suffering.”*; (2)*” Any action that causes pain and discomfort to the animals.”*
CI 4 Movement restriction	3 (5.7%)	*43 (14.2%)*	6.92	0.0084	(1)“…to restrain the movement…”; (2) “…imprisonment of the animal…” (1) *“Any confined animal is mistreated.”* (2) *“To leave animals confined...”*
CI 5 Deviation from naturalness	5 (9.6%)	*13 (4.3%)*	0.56	0.5395	(1)“… practices that move them away from their natural living condition.”; (2) “...not to be in an aproppriate environment for the species.” (1) “Impossibility to express natural behaviors…” (2) *“…to raise an animal isolated from other members of the species…”*
CI 6 Abandonment	-	*21 (6.9%)*	6.00	0.0220	(1) *“Abandonment.”*
CI 7 Emotional neglect	-	*11 (3.6%)*	3.02	0.1200	(1) *“From lack of affection to aggression”. (2) “...no attention and no affection.”*
CI 8 Non-classifiable	5 (9.6%)	*8 (2.6%)*	2.58	0.1557	(1)“ Dogs do not always cause mistreatment, sometimes they help.” (2)“However, I believe that animal maltreatment has considerably decreased due to so many new( technique)s.” (1) *“The word maltreatment says everything.”*

^1^ A statement might be classified in one or more CI.

**Table 5 animals-11-01903-t005:** The level of relevance on a five-point scale, from 1 (not relevant) to 5 (very relevant), regarding animal maltreatment debates and reasons given by citizens from Porto Alegre, South Brazil, from September 2017 to December 2017.

Level of Relevance (*n* = 168)	Categories of Reasons ^1^	Number of Quotes ^2^ (%)	Examples of Quotes (*n* = 105) ^3^
Slightly relevant (*n* = 2; 1%) or not relevant (*n* = 0)	Animal welfare is slightly important	2 (100)	(1) “There are other issues that I believe are more important nowadays. For instance, the efforts for abortion liberation; although I perceived animal welfare as important, I believe that killing a defenceless human being is much worse than mistreating an animal.” (2) “Because there is a social inversion of values”
Moderately relevant (*n* = 15; 9%)	Independent reasons	7 (100)	(1) “All polemic issues must be largely discussed” (2) “a conscientious discussion is valid, but many times what we see are personal arguments with no basis on the reality in the field.”
Relevant (*n* = 25; 15%) or very relevant (*n* = 126; 75%)	Due to ethical issues related to animals	38 (50.0)	(1) “The way we treat animals reflects our empathy even for ourselves as humans.” (2) “A society only evolves with respect to life in all its expressions.”
	Because animals are living beings and sentience	26 (34.2)	(1) “Because all that relates to life is relevant.” (2) “Because it is time for us to evolve as humanity and to see other live beings as beings with rights and that feel pain.”
	To understand the context and to propose a solution	26 (34.2)	(1) “Animal maltreatment debates may reveal situations which are unknown to society, and as it is known, it is possible to interfere on the process to eliminate it.” (2) “The discussion may result in alternatives to avoid maltreatment”
	It is a crime	6 (7.9)	(1) “I think it is important to press charges against animal maltreatment” (2) “Arrest is a must.”
	Not classified	6 (7.9)	(1) “Obvious answer.” (2) “There is nothing to explain.”

^1^ Quotes from statements were classified into categories of reasons; ^2^ A statement might be classified in one or more category of reasons; ^3^ From 168 answers regarding the level of agreement, 105 respondents left comments.

**Table 6 animals-11-01903-t006:** The level of agreement on a five-point scale, from definitely yes to definitely no, regarding the intention to purchase food from animals who were maltreated and reasons given by citizens of Porto Alegre, South Brazil, from September to December 2017.

Level of Concordance (*n* = 167)	Categories of Reasons ^1^	Number of Quotes (%)	Examples of Quotes (*n* = 89)
Definitely yes (*n* = 9; 5%) or yes^4^ (*n* = 9; 5%)	Non-classifiable	3 (100%)	(1) “I know it is not right, but in the end many times I overlook this factor.” (2) “I would buy because I do not have much money…”
Neutral (*n* = 18; 11%)	Lack of options on the market	4 (50.0%)	(1) “If I were hungry and this were the only option, I would purchase it. But, if I can choose, I would never purchase it.” (2) “Because sometimes there is no other option.”
	Cultural reasons	3 (37.5%)	(1) “It is hard because we are culturally used to nutritional habits involving a diversity of animals. However, maybe more reflexion on my side about this is needed.” (2) “Something to consider is the local culture.”
	Non-classifiable	1 (12.5%)	(1) “Depends on what is considered maltreatment.”
No (*n* = 21; 13%) or definetely no (*n* = 93; 66%)	If option or information were available	9 (9.3%)	(1) “If I knew this fact, I would never purchase it.” (2) “It is possible that I would have already adopted such a posture if I had sufficient information on how these animals are treated.”
	Concerns towards people	9 (9.3%)	(1) “Because probably such animals would not be healthy.” (2) “Because certainly the product would be of poor quality.”
	Not to contribute to companies using bad practices/to boycott these companies	24 (24.7%)	(1) “Because it is a way to fund the violence against animals.” (2) “Because if the consumer imposes restriction, the food chain will be obliged to reformulate their attitudes.”
	Concerns towards animals	44 (45.3%)	(1) “Because of the suffering to which they were exposed” (2) “I don’t think it is right to outsource the suffering to feel free from guilt.”
	Non-classifiable	11 (11.3%)	(1) “Is there a need to justify it?” (2) “Christian duty.”

^1^ Quotes from statements were classified in the category of reasons.

## Data Availability

Original responses to our questionnaires are available in Portuguese, upon request.

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
