# Peer review of "Mind the Gap: Animal Protection Law and Opinion of Sheep Farmers and Lay Citizens Regarding Animal Maltreatment in Sheep Farming in Southern Brazil"

_animals, 2021, doi:10.3390/ani11071903_

Round 1

Reviewer 1 Report

A very good paper.

Please include the ethical review permissions in your methodology

Reviewer 2 Report

Dear authors, congratulations on this well-written publication. The introduction clearly describes the research topic, the research methodology is clearly described, the results follow the outcome and the discussion session nicely goes over the different results. The statistics are also well done. The publication is well referenced with a good seclection of references. What I especially like is that the study very well describes the disclaimer as the citizen's answering the survey did not represent average citizens, being higher educated and a higher number of females. 

The only comments I can make on the publication are really minor ones, namely: 

  • line 123 - extra space between September and to
  • line 235 - the whole publication is long, with the most interesting part being the discussion. I wonder if it would not help to make it a bit shorter e.g. the part on the Latent Class Analysis personally I do not think adds a lot and could be cut. 

Reviewer 3 Report

  1. This is a very important study. As someone who teaches animal law – the issues identified by this study are critical  - ie the concept of animal welfare or cruelty, varies according to the situation.
  2. I am assuming that the researchers applied for ethics clearance for the study.
  3. The summary and the abstract use the word “maltreatment”, (also from line 65), however in the introduction from line 53 the authors use the word “welfare”. Two points about this: 1. Maltreatment and welfare are not necessarily the same (to be cynical, the latter depends on the level of maltreatment that society and legal systems deem acceptable); 2. If the word welfare is used for the article, it needs a definition.  This is from the OIE terrestrial code (2019 – may need updating) “the physical and mental state of an animal in relation to the conditions in which it lives and dies. An animal experiences good welfare if the animal is healthy, comfortable, well nourished, safe, is not suffering from unpleasant states such as pain, fear and distress and is able to express behaviours that are important for its physical and mental state”.  OIE, Terrestrial Animal Health Code, Article 7.1.1, 28th edition (2019) available from https://www.oie.int/index.php?id=169&L=0&htmfile=chapitre_aw_introduction.htm.
  4. If the authors wish – this is the citation for the OIE International Agreement for the Creation at Paris of an International Office for Dealing with Contagious Diseases of Animals and Annex, opened for signature 25 January 1924, [1925] ATS 15, (entered into force 12 January 1925). As at June 2021 the organisation had 182 members.  Although they may wish to cite it to something other than the Australian Treaty Series.  The point to this long-winded parag is that it would be helpful to distinguish between animal welfare and cruelty/maltreatment.
  5. Line 59 is very broad – needs a reference if possible… not sure whether this is of help https://vaci.voiceless.org.au/about-the-vaci/
  6. Line 88ish – not sure I understand what is meant by vulnerability in this context
  7. The methodology was explained very clearly – but line 235 – I am not a statistician (my research is mainly qualitative) and did not understand Latent Class Analysis. There may be readers who are interested in this article in the same position as I am. This is an important part of the project – perhaps it could be explained in plainer English (is it a control group?)
  8. Lines 401- 402 say “more sensitive perception of animal maltreatment than farmers,” – this is another reason that terms need to be defined.  Is it sensitive or perhaps “nuanced”? Especially given the conclusion in lines 431-432. Anyway – up to the researchers.
  9. Lines 512-514 “Thus, it is possible that farmers were less sensitive than citizens to movement restriction as they may have linked it with beneficial situations for the animals” – is that for the animals themselves or for the animals as commodities in production.
  10. Line 625 “to practice” should be “to practise” – in this usage the word is a verb
